# Long-Term Persistence of Mitochondrial DNA Instability in HIV-Exposed Uninfected Children during and after Exposure to Antiretroviral Drugs and HIV

**DOI:** 10.3390/biomedicines10081786

**Published:** 2022-07-25

**Authors:** Valérie Desquiret-Dumas, Morgana D’Ottavi, Audrey Monnin, David Goudenège, Nicolas Méda, Amélie Vizeneux, Chipepo Kankasa, Thorkild Tylleskar, Céline Bris, Vincent Procaccio, Nicolas Nagot, Philippe Van de Perre, Pascal Reynier, Jean-Pierre Molès

**Affiliations:** 1Department of Biochemistry and Genetics, University Hospital of Angers, F-49000 Angers, France; vadesquiret@chu-angers.fr (V.D.-D.); david.goudenege@chu-angers.fr (D.G.); cebris@chu-angers.fr (C.B.); viprocaccio@chu-angers.fr (V.P.); pareynier@chu-angers.fr (P.R.); 2UMR MITOVASC, CNRS 6015, INSERM U1083, University of Angers, F-49000 Angers, France; 3Pathogenesis and Control of Chronic and Emerging Infections, INSERM, Etablissement Français du Sang, University of Montpellier, University of Antilles, F-34394 Montpellier, France; morgana.d-ottavi@umontpellier.fr (M.D.); audrey-a91@hotmail.fr (A.M.); amelie.vizeneux@gmail.com (A.V.); n-nagot@chu-montpellier.fr (N.N.); p-van_de_perre@chu-montpellier.fr (P.V.d.P.); 4Centre MURAZ, Bobo-Dioulasso 01 B.P. 390, Burkina Faso; meda_nicolas@yahoo.fr; 5Department of Paediatrics and Child Health, University Teaching Hospital, Lusaka P.O. Box 50001, Zambia; ckankasa@uthhap.org.zm; 6Centre for International Health, University of Bergen, N-5020 Bergen, Norway; thorkild.tylleskar@uib.no

**Keywords:** HIV, neonates, mitochondria, clinical outcomes, neurodevelopment, genetic alterations

## Abstract

HIV-exposed uninfected (HEU) children show impaired health outcomes during childhood. A high rate of mitochondrial DNA (mtDNA) instability was reported in the blood of HEU at birth. We aimed to explore the relationship between these health outcomes and mtDNA deletions over time in a case series of 24 HEU children. MtDNA instability was assessed by deep sequencing and analyzed by eKLIPse-v2 algorithm at three time points, namely birth, 1 year, and 6 years of age. Association between mtDNA deletion and health outcomes, including growth, clinical, and neurodevelopmental parameters, were explored using univariate statistical analyses and after stratification with relevant variables. HEU children were selected with an equal male:female ratio. An elevated number of mtDNA deletions and duplications events was observed at 7 days’ post-partum. Median heteroplasmy increased at one year of life and then returned to baseline by six years of age. The mtDNA instability was acquired and was not transmitted by the mother. No risk factors were significantly associated with mtDNA instability. In this small case series, we did not detect any association between any health outcome at 6 years and mtDNA instability measures. A significant effect modification of the association between the duration of maternal prophylaxis and child growth was observed after stratification with heteroplasmy rate. Genomic instability persists over time among HEU children but, despite its extension, stays subclinical at six years.

## 1. Introduction

Mitochondria are a metabolic hub involved in cellular respiration, energy production, biosyntheses, redox equilibrium, and signaling [1]. Their activities and number per cell are tightly regulated through a cross-talk with the nucleus, which encodes regulatory networks and most of their components [2]. They contain their own DNA (mtDNA) of exclusive maternal origin which encodes for 13 subunits of the respiratory chain [1]. As compared with nucleic DNA, mtDNA is more susceptible to genetic alterations, as mitochondria harbor inefficient DNA repair systems [3,4].

Mitochondrial cytopathies are a heterogeneous group of disorders sharing a primary defect of the mitochondrial respiratory chain [5]. Some of these cytopathies are due to mtDNA mutations transmitted maternally (point mutations) or appearing sporadically (single deletions) [6]. Others cytopathies with a Mendelian inheritance are related to multiple mtDNA deletions secondary to mutations in nuclear genes involved in mtDNA maintenance, such as polymerase gamma (POLG). Finally, many other nuclear genes, disrupting various mitochondrial functions when mutated, are also responsible for a large set of mitochondrial cytopathies [6].

In addition to these inherited diseases, mtDNA instability, i.e., an accumulation of point mutations or large scale rearrangements, is observed in cancer cells, degenerative diseases, and, more generally, in physiological aging [2]. In this case, the levels of mtDNA mutants are generally lower than those observed in mitochondrial cytopathies but with multiple variants cumulating their effects. This accumulation of mtDNA mutations is mainly observed in post-mitotic cells or tissues, where the natural selection of cells with the best mitochondrial heritage is limited. Thus, if the accumulation of mtDNA mutations is frequently observed in skeletal muscle or brain, it is rarely observed in blood cells whose renewal rate allows such selection [7]. The phenotypic consequences of these heteroplasmic variants are difficult to evaluate and to predict, a functional alteration of the respiratory chain requiring several tens of percent, and a clinical impairment of probably more than 50% of the mutant rate, or even higher levels [8]. High-throughput sequencing has greatly improved the exhaustive exploration of mtDNA, with a reading depth allowing for the detection of low levels of heteroplasmy, and thus a more sensitive exploration of mtDNA instability. In addition, computational pipelines of deep sequencing data have allowed for much more exhaustive cartography at the nucleotide level of mtDNA deletions and duplications [9,10,11,12].

In the fight against the HIV epidemic, one of the most important achievements has been the prevention of mother-to-child transmission [13]. Albeit not complete with an estimated 150,000 new pediatric infections worldwide in 2020, the result of these programs is the birth of about 1 million children exposed to HIV in utero but uninfected (HEU) each year [14]. The children have serious negative health outcomes compared to HIV-unexposed uninfected children (HUU), including premature birth, impaired growth, neurodevelopment and immune response, and high mortality [15,16]. Reasons are likely multifactorial and multitemporal, with exposures to HIV and antiretroviral (ARV) drugs in utero, during labor/delivery and post-partum during breastfeeding, or through infant prophylaxis, as well as through the impact of being fostered by an HIV-infected mother [16]. Such broad complications would match well with a ubiquitous dysfunction such as mitochondrial dysfunctions, which was in fact already reported as a consequence of some types of ARV treatment [17,18]. Among HEU, the number of mtDNA copies per cell, a proxy of mitochondrial fitness, was investigated for many years. It is difficult to draw definitive conclusions because exposures to ARV treatment or to HIV viral load as well as other covariates such as maternal smoking are too diverse [19,20,21]. If the quantitative evaluation of the mtDNA copy number was addressed in several contexts, qualitative investigations of mtDNA are scarce and focus on the most common deletion [22,23,24,25,26] or the mutation rate at the nucleotide level [22,23,27]. We recently applied the deep sequencing approach coupled with an eKLIPse computational pipeline to HEU at birth and reported that 75% of the HEU have deleted mtDNA molecules in blood cells, which is not found in control HUU at birth [28]. The heteroplasmy rates ranged from 0.01% to up to 50%. Most of them were subclinical rates, though substantial enough for raising concerns regarding their evolution over time.

Primary objectives were to explore the mtDNA alteration pattern of HEU children during their exposure to HIV and ARV and years after the discontinuation of these exposures, to differentiate deletion from duplication events, and to identify the associated risk factors and potential association with health outcomes later in life.

## 2. Materials and Methods

### 2.1. Study Design

This study is a case series describing and analyzing the short-term and long-term effects of extended ARV prophylaxis among HEU on mtDNA genetic instability. Time points were “before the prophylaxis” (Day 7, D7), “at the end of the prophylaxis” (Week 50, W50), and “five years’ post-prophylaxis” (Year 6, Y6).

### 2.2. Study Population

Twenty-four children were selected from two previous studies describing mtDNA outcomes among HEU [21,29]. These were infants born to HIV-infected mothers from Burkina Faso and Zambia and received six days of nevirapine from birth followed by one year of lamivudine or lopinavir/ritonavir extended prophylaxis. Children were randomly assigned to one of the two arms at day 7 as per PROMISE-PEP protocol (NCT00640263). In Zambia, circulating HIV viruses were of clade C and in Burkina Faso of subtype CRF2 and CRF06. Subsequently, they were enrolled in the PROMISE M&S study (NCT03519503), which aimed to evaluate the growth, clinical, and neurodevelopment outcomes at school age in these children. These trials were conducted between 2009 and 2012, and again from 2017 to 2018, in four African countries: Burkina Faso, South Africa, Uganda, and Zambia. A detailed sample selection flow is given in Appendix A Appendix A. All demographic, anthropometric, and clinical data were obtained from both the PROMISE-PEP and the PROMISE M&S trial databases [30,31].

### 2.3. Sample Collection and Processing

Children’s whole capillary blood was collected at D7, W50, and Y6 by heel or finger prick and directly processed on dried blood spots (DBS, Whatman 903 cards, GE Healthcare, Cardiff, UK). DBS were stored in a zipped pouch with desiccant at −20 °C until their shipment to France. Then, DNA was extracted using the QIAamp DNA Blood Mini Kit (Qiagen, Hilden, Germany), following the manufacturer’s instructions, from 3 mm diameter punches (*n* = 3) and stored at −80 °C. Mother’s PBMCs were prepared from the Ficoll–plasma interface, washed three times in PBS/2% Fetal Bovine Serum, and stored as dry cell pellet at −80 °C.

### 2.4. Detection and Quantification of Mitochondrial DNA Instability: The eKlispe-v2 High-Throughput Computational Pipeline

The assay combined the deep high-throughput sequencing of mtDNA and the bioinformatic pipeline [11]. The sample flow and the quality controls were described in detail in Monnin et al., 2021 [28]. Briefly, long-range mtDNA PCR (2 overlapping fragments) was performed from 25 ng of total blood DNA. The PCR products were next fragmented using an enzymatic fragmentation approach and the library constructed using the AB Library Builder system (Ion Xpress™ Plus Fragment Library Kit, Thermo Fisher Scientific, Waltham, MA, USA). Emulsion PCR was then performed using Ion Chef apparatus (Thermo Fisher Scientific, Waltham, MA, USA), and DNA fragments were sequenced on the Ion S5-XL on Ion 540 chips (Thermo Fisher Scientific, Waltham, MA, USA). Run parameters met the manufacturer recommendations, and the mean number of reads per base ± SD for all sample was of 9424 ± 4640, and there was no contamination with the negative control sample. The criteria to validate a deletion or a duplication event consisted in: (i) a minimal sequencing depth of 100 reads per base, (ii) a minimal number of identical 10 reads, and (iii) an heteroplasmy rate of 0.05%. The bioinformatics pipeline was modified since Monnin et al., 2021. We used the eKLIPse version-2 algorithm, which differs from the version-1 by the python version used (python 2 vs. 3) and the exclusion of artefactual breakpoints. It also discriminates between deletion and duplication events based on the fact that deleted mtDNA molecules without origins of replication cannot exist and are therefore necessarily duplicated regions inserted in an mtDNA molecule having these origins of replication [11].

### 2.5. Statistical Analysis

Participant characteristics are described using means with standard deviations (SD) or medians with interquartile range (IQR) for continuous variables and percentages for categorical variables. To assess potential selection bias in our study subset, baseline characteristics were also compared to the characteristics of the other randomly selected HEU that were not included in the analysis.

Anthropometric Z-scores (weight-for-age, height-for-age, weight-for-height, and body mass index) were calculated using the 2006 WHO child growth standards [32]. The Z-scores for systolic blood pressure were calculated using the formulas and parameters provided by the MSD Manual [33]. Hematological and biochemical results were categorized based on the DAIDS tables for grading the severity of adult and pediatric severe adverse events [34]. Neuropsychological global performance scoring methodology has been previously described elsewhere [31].

We first investigated the association between an elevated cumulative heteroplasmy rate (dichotomized using a threshold set at 1%) at D7 and the following potential risk factors: study site, sex, gestational age, parity, mother’s age at birth, mother education, mother HIV viral load at D7, duration of ARV prophylaxis during pregnancy, and alcohol consumption during pregnancy. This threshold is above the limit of quantification for the heteroplasmy rate set at 0.5% [35]. Fischer’s exact test of independence was performed for categorical variables. For continuous variables, Spearman’s rho coefficient was calculated using cumulative heteroplasmy as a continuous variable.

Second, we addressed the association between the child’s ARV prophylaxis regimen given at birth (lopinavir/ritonavir (LPV/r) or lamivudine (3TC)) and the following outcomes at W50: the total number of mtDNA alteration events, cumulative heteroplasmy rate, the total number of mtDNA deletion events, and the total number of mtDNA duplication events. We compared outcomes between prophylaxis regimens using a mixed effects linear regression model for repeated measures with the Kenward and Roger degrees-of-freedom adjustment for standard error calculations. Results are presented as regression coefficients with their 95% confidence intervals.

We then explored the association between an elevated cumulative heteroplasmy rate (>1%) at D7 and indicators of the child’s health at Y6. For this purpose, we assessed the association with growth outcomes (head circumference, weight-for-age Z-score, height-for-age Z-score, body mass index Z-score, and systolic blood pressure Z-score), clinical outcomes (presence of any abnormalities, hospital admissions since the first year of life, LDH and ALT concentrations), and neuropsychological outcomes (global performance scores obtained from the 10questionsPlus questionnaire; SDQ-25, Strengths and Difficulties Questionnaire; TOVA, Test Of Variable of Attention; MABC-2, Movement Assessment Battery for Children second edition; and KABC-II, Kaufman Assessment Battery for Children second edition). We compared outcomes according to the cumulative heteroplasmy rate using Student’s t test or Fischer’s exact test for continuous or categorical variables, respectively.

Finally, we assessed the association between growth and clinical outcomes at Y6 and three predictive factors: duration of maternal prophylaxis during pregnancy, maternal viral load at birth, and child weight-for-height Z score at birth. This association was then stratified by the infants’ heteroplasmy rate at birth in order to evaluate the possibility of effect modification. Associations are presented as linear regression coefficients and their 95% confidence interval.

None of the models were adjusted for potentially confounding factors as this analysis is exploratory and based on a small sample of only 24 HEU. The threshold for statistical significance was set at α < 0.05. Statistical analyses were performed using Stata 16.1(Copyright © 2022–2019, StataCorp LLC, College Station, TX, USA).

## 3. Results

### 3.1. Study Population Characteristics at D7, W50, and Y6

Half of the selected HEU children were girls, half received oral 3TC as prophylaxis during the first year of life. Their country of residence was distributed almost equally between Burkina Faso and Zambia (Table 1). At D7, anthropometric and clinical variables were within the normal ranges. Mothers were not eligible to receive ART treatment at the time of the trial as per WHO’s recommendations; however, to prevent perinatal HIV transmission, all mothers received an antenatal AZT-containing regimen for a median duration of eight weeks. Overall, forty one percent of mothers had an unsuppressed HIV viral load (Table 1). These selected children had the same characteristic as those of the country of origin, except that included mothers were more likely to have consumed alcohol during their pregnancy compared to those that were not included (*p* = 0.013, Appendix A Appendix A). During the follow-up, neutropenia was detected at W38 post-partum in 18% of children, and 6% had raised ALT levels in blood. At Y6, growth values were within the normal ranges. However, most HEU had abnormal blood pressures. During the clinical examination, four (17%) of the children had one organ abnormality: blindness, hearing impairment, mild arrhythmia, and heart murmur (Table 2). One-fifth had been hospitalized at least once since their first year of life, 20% had a grade 2 abnormal LDH concentration, and none had abnormal ALT concentration.

### 3.2. MtDNA Alterations among HEU

Out of the 24 samples collected at D7, 22 had been previously analyzed with eKLIPse version-1 algorithm [28], and analyses were repeated with the new version of the software. The percentage of children with mtDNA alterations was similar regardless the version of eKLIPse. Overall, 119 events of mtDNA alterations were recorded at the three time points (Table 3). The prevalence of mtDNA alteration events was high among HEU at D7, increased at W50, and declined to baseline values at Y6 (see example in Figure 1A). All but one of the children with alteration events at D7 had alterations at W50. Few children acquired alteration events at W50 (*n* = 7) and at Y6 (*n* = 1) (see Appendix A Appendix A). The deletion/duplication ratio remained similar throughout follow up, deletion being more frequent, from 55% at D7 to 69% at Y6. Only six events persisted at two time points. The regions targeted by these genetic alterations were different for deletions and duplications (Figure 1B,C). Out of the 76 deletion events, 19 (25%) consisted in a 7 kb deletion between nucleotide position 7034 and 14,400. Out of the 43 duplication events, 9 (21%) consisted in a duplication of the region from nucleotide position 14,857 to 6891 (data not shown). Regarding the deletion events, three classes are described according to the presence or absence of the same sequence flanking the deleted region. The distribution of the class I/II/III did not change neither over time (*p* = 0.555, data not shown) nor according to child prophylactic regimen. Median heteroplasmy increased at W50 of prophylaxis and then dropped back to its baseline value at Y6. However, the minimum and maximum heteroplasmy values ranged from 0.06% to 99.7% at D7, from 0.2% to 57.7% at W50, and from 0.1% to 3.17% at Y6 (Table 3). The 99.7% heteroplasmy rate for one child consisted in a heteroplasmy deletion rate of 84.9% and a heteroplasmy duplication rate of 15.8%. Finally, we did not report any pathogenic point mutation at the three time points.

When stratified by the infants’ type of prophylaxis regimen, the proportion of infants with mtDNA alterations was similar at W50. Infants having received one year of LPV/r prophylaxis showed an increased median heteroplasmy rate at W50, from 0.08% [0.00–0.59] at D7 to 0.54% [0.25–2.27] at W50, as compared to infants having received 3TC prophylaxis who remained stable, from 0.41% [0.11–3.21] at D7 to 0.32% [0.16–2.05] at W50 (Table 3). The proportion of children with a heteroplasmy rate above 1% at Y6 was equivalent in the two groups (data not shown).

### 3.3. Determinants of mtDNA Instability at D7

None of the tested determinants were statistically associated with an elevated cumulative heteroplasmy rate–set as above 1% (Table 4). 

### 3.4. Impact of the Prophylactic Drug Exposure on mtDNA Instability

At W50, children exposed from D7 to W50 to either the LPV/r or 3TC drug had a comparable number and quantity of mtDNA alteration events (Figure 2).

### 3.5. Comparison of Clinical and Biological Characteristics at Y6 by mtDNA Heteroplasmy Rate Early in Life

Detection of a cumulative heteroplasmy rate greater than 1% at birth was not associated with growth abnormalities, disease prevalence, or impaired neurodevelopment (Table 5). None of the variables were statistically associated; however, the average WAZ tended to be lower, and the percentage of children having been hospitalized between W50 and Y6 tended to be higher for those children that had an elevated heteroplasmy rate at birth (*p* = 0.142 and *p* = 0.126, respectively; Table 5). Furthermore, HEU with a heteroplasmy rate greater than 1% at either birth or W50, on average, tended to have lower WAZ, BMIZ, and KABC-II scores (*p* = 0.111, *p* = 0.104, and *p* = 0.130 respectively, Appendix A Appendix A). Growth, hematological, and biochemistry outcomes were not associated with high heteroplasmy rates at W50 (data not shown).

Poor growth and clinical outcomes at 6 years of age were not significantly associated with the duration of maternal prophylaxis during pregnancy, maternal viral load at birth, or child WHZ at D7. However, when stratified by the infants’ heteroplasmy rate at birth, we observe a statistically significant potential effect modification of the association between the duration of maternal prophylaxis and child head circumference at 6 years of age (*p*-value for interaction = 0.033; Table 6). Among children who had an elevated heteroplasmy rate at birth, average head circumference significantly increased by 0.31 cm [0.06–0.55] for each additional week of in utero ARV exposure. When stratified by the infants’ heteroplasmy rate at birth, we also observe a statistically significant potential effect modification of the association between the duration of maternal prophylaxis and child BMI Z-score at 6 years of age (*p*-value for interaction = 0.047; Table 6). On average among children who had an elevated heteroplasmy rate at birth, BMI Z-scores significantly decreased by −0.34 units [−0.66–−0.02] per additional week of in utero ARV exposure, whereas there appears to be little to no change in BMI Z-scores according to maternal prophylaxis duration among children whose heteroplasmy rate at birth was ≤1%. Moreover, though not quite statistically significant, it is of note that, among children who had an elevated heteroplasmy rate at birth, ALT concentration increased with longer exposure to maternal ARV, while it was almost null among children whose heteroplasmy rate at birth was ≤1% (*p* for interaction = 0.070; Table 6). No interactions were observed between elevated heteroplasmy and maternal viral load or infant WHZ at birth as predictors for poor health outcomes at 6 years of age (Appendix A Appendix A). Moreover, of note, though not statistically significant, is that, before stratification and among children whose heteroplasmy rate at birth was ≤1%, LDH concentration at Y6 tended to decrease with longer in utero ARV exposure; however, among children who had an elevated heteroplasmy rate at birth, LDH concentration increased on average by 2.87 IU [−2.3–8.05] per additional week of in utero ARV exposure (*p* for interaction = 0.255; Table 6).

## 4. Discussion

Benefiting from a unique blood sample collection, allowing the follow-up of HEU children from birth up to six years of age, as well as from the progress of deep mtDNA sequencing, and we here demonstrate that mtDNA instability observed at birth persisted for many years in children after HIV and ARV drug exposure. Significant levels (>1%) of these mtDNA deletions and duplications were observed in 29.2% of children at day 7, 37.5% at week 50, and 12.5% at year 6. Median heteroplasmy levels at birth (0.24%) increased after one year of prophylaxis (0.42%) and then fell back to their birth level at year 6 (0.24%). Two-thirds of the events were deletions and the remainders were duplications, without any point mutation detected.

The children in the present study were followed up to six years of age with detailed clinical investigations, including on neurodevelopment. HEU children were previously described as having growth impairments, cardiac abnormalities, and neurodevelopment defaults [16]. All of these clinical outcomes could result from mitochondrial dysfunctions. In this small open series, these events did not appear to be associated with any clinical observations at six years of age. Nevertheless, in utero ARV drug exposure in conjunction with a heteroplasmy rate >1% appeared to have impacted several growth and biological variables. Among systemic biochemical markers of mitochondrial syndromes, ALT concentrations were within normal limits, whereas LDH concentrations were high in almost two-thirds of the children. 

The first described mtDNA deletion, namely the common ∆4977bp mtDNA deletion, accumulates according to age, especially in post-mitotic tissues [36,37]; however, so far, only a few studies have analyzed mtDNA deletion data via a deep sequencing approach by age group [11,12]. These studies also showed an increasing proportion of mtDNA rearrangements with age as well as with carcinogenesis [8]. Bris et al. reported using the same computational pipeline for an average of 0.18% deletion rate in the uroepithelial cells of healthy persons aged 50 years and above [11]. Uroepithelial cells are commonly used in our diagnostic activities for mitochondrial diseases because they usually express higher levels of heteroplasmy of pathogenic variants [38]. Our results showed levels of mtDNA rearrangements in blood HEU similar to those found in urine from a population over 50 years old, suggesting early mtDNA instability in blood cells. However, as blood cells are less prone to the accumulation of mtDNA rearrangements, these significant but low levels of heteroplasmy raise concerns that higher levels of heteroplasmy may be present in post-mitotic tissues such as the nervous system and muscle with potential deleterious clinical consequences. Furthermore, the long-lasting detection of such instability, up to six years of age, together with the rapid renewal of blood cells and the discontinuation of the HIV and/or ARV exposure, suggest that these alterations occur also in blood stem cells. It also suggests that the replicative advantage of the deleted molecules outweighed their physiological elimination processes [39,40]. It is noteworthy that, even if no threshold was definitively set to determine a clinical relevance for these mtDNA alterations, it has been demonstrated that a cell with a heteroplasmy above 55% is impaired in its mitochondrial functions [8]. However, lower levels of heteroplasmy can be observed in patients with mitochondrial diseases [5]. The mtDNA duplication phenomenon has only recently been identified [10], so far with no demonstrated impact on the mitochondrial function. In our study, about a quarter of mtDNA instability corresponded to one deletion (∆7034 nt/14,400 nt) and one duplication (∆14,857 nt to 6891 nt), most probably resulting from the unique inducers, which are yet to be identified.

MtDNA instability was previously reported in an HIV context with ARV treatments as inducers [41,42]. For HEU children, factors inducing such DNA instability remain elusive. First, for three quarters of the children, deletions differed from one child to the other and overtime in each of them. This observation is in accordance with the fact that no deletion was detected among the paired HIV-infected mothers, suggesting that deletions are not vertically transmitted. If acquired, both HIV and ARV drug exposures might be considered. While ARV exposure can be characterized as a type and dose of ARV and duration of exposure, HIV exposure is more complex and more difficult to estimate. Such exposure includes items such as exposure to free HIV proteins already known as mitotoxicant [43], but also the known microinflammation of all HIV-infected patients or their altered metabolism. Deciphering between those three and others is not yet technically possible. Indeed, at D7, neonates have already been exposed to maternal HIV status and to ARV drugs from both the maternal and their own ARV prophylaxis. By their first year of life, children cumulated the same exposure, i.e., the maternal HIV status, through breastmilk and LPV/r or 3TC drugs. By year six, these exposures had been discontinued for the last five years. Mechanisms exist within a cell to eliminate defective mitochondria [39,40]. These mechanisms are active up to a certain threshold, after which the cells undergo cell death. The threshold is not defined, and may vary from one cell type to another and most probably with age, neonates, and the elderly being less effective. The rate of clearance of these genetic alterations is also unknown and may also vary by cell type and age. Nonetheless, ARV drug exposure during in utero life was evaluated in terms of mitochondrial toxicity among *Patas* monkeys at distance of birth. Noticeably, at three years of age, mitochondrial toxicity was still detectable, suggesting that the rate of elimination after exposure discontinuation was slow [44,45]. This observation in monkeys suggests that later clinical manifestation or early ageing cannot be ruled out in the participants of our study with the more severe profile of mitochondrial defects. The child prophylaxis taken during the first year of life did not worsen the genetic alteration status in this open series. However, we noticed a selection bias, as baseline values for 3TC or LPV/r prophylaxis were different and LPV/r regimen increased the level of heteroplasmy, whereas, upon 3TC regimen, heteroplasmy was levelled (Table 3). The proportion of deletion versus duplication and the classes of deletions were equivalent across prophylaxis regimens. Recently, numerous biological outcomes have been evaluated among persons initiating post-exposure prophylaxis after a non-occupational sexual exposure to HIV [46]. Authors have reported that one-month regimens containing AZT were the most damaging in term of mitochondrial toxicity. Currently, extended child prophylaxis is added to the prevention of mother-to-child HIV transmission guidelines in many countries that continue to face an unacceptably high rate of pediatric HIV infections. Some of these regimens, such as that recommended by the Zambian national guidelines, consist of a three-drug regimen including AZT, 3TC, and nevirapine [47]. It is noteworthy that all of the HEU of this short case series were also exposed to maternal AZT treatment during pregnancy. Therefore, we cannot definitively rule out that the observed mtDNA instability in HEU results from the in utero maternal AZT treatment, similarly to what was reported in *Patas* monkey experiments. Analyzing a group of mothers not receiving treatment is needed. Polyexposure to mitotoxic drugs deserves a new comprehensive investigation scheme. Similarly, in HIV-infected children, adding mito-protective drugs to ARV therapy could be considered [48,49].

This study has several limitations. The cohort of children analyzed is small but gave us the opportunity for the long-term follow-up of patients at three time points. The small number of subjects hinders us from testing the combination of various variables of interest and adjusting the associations for known confounding variables. The small sample size also hinders our ability to draw conclusive results from any of the exploratory analyses that were done on this sample. The results presented herein should be considered as an insight into the possible nature of the relationship between parameters; however, further investigation using a larger sample size and a control group is necessary for establishing any conclusive association or effect. Secondly, the PCR design amplified the circular mtDNA into two amplicons, and deletions within the regions targeted by the primers could not be identified. Lastly, we did not have the opportunity to test HUU to compare our observations to an HIV-unexposed control group. However, we previously analyzed DBS collected during the same period from healthy French newborns and found no mtDNA deletion [28]. We have previously showed that neither the use of DBS nor the duration of storage introduced artefactual deletions. Of note, in this DBS validation study, half of the participants had an African mtDNA haplotype [28]. From the diagnostic laboratory archive (National Reference Center for Mitochondrial Diseases, University Angers, France), we were able to retrieve the results of 26 eKLIPse assays performed on the blood of children under ten years of age for whom mitochondrial disease was suspected. One deletion was identified with a heteroplasmy of 0.1%, giving a prevalence of 3.8%. We also retrieved analyses performed on urine, the bodily fluid recommended as the first choice for primary investigation of mitochondrial diseases thanks to a greater sensitivity than blood [38]. Out of 135 urine samples, only one sample from a child under ten years of age had one mtDNA deletion with a high heteroplasmy (69%). This patient was later diagnosed with Kearns–Sayre syndrome. Together, these figures strongly support the notion that deletion events are very rare in children, and that the presence of these rearrangements in the blood of HEU patients is quite unusual.

Overall, if mtDNA alterations were detected in the blood of HEU children from birth to six years, they did not appear to be associated with poor health outcomes. Such genomic instability is, however, not reassuring, as it indicates possible early mtDNA instability among HEU children, and that they raise concerns of much higher levels of heteroplasmy in organs such as the brain or muscles.

## Figures and Tables

**Figure 1 biomedicines-10-01786-f001:**
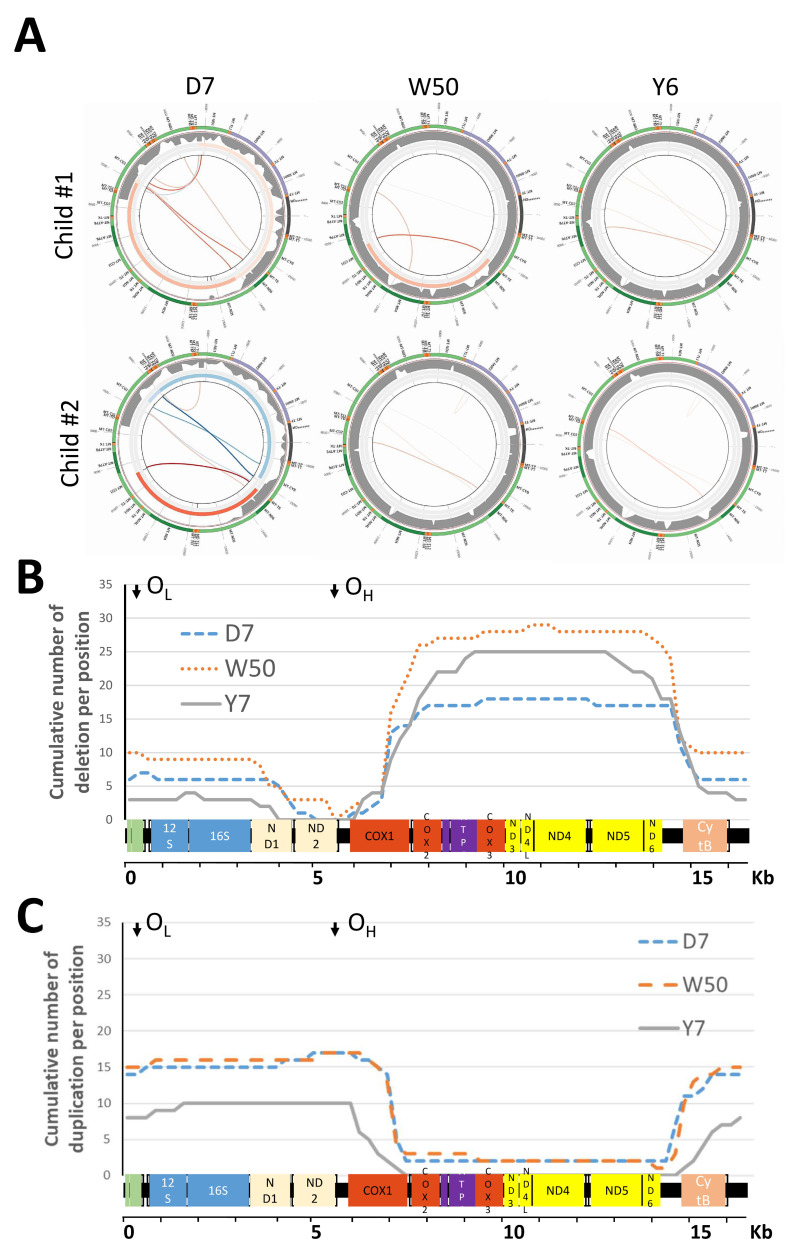
mtDNA deletions among HEU children. Subfigure **A** shows schematic representation of the eKlipse high-throughput computational pipeline result. Three mitocircles are shown per child at the three time points. The outer circle represents the mtDNA genome. Deletions are visualized in the center of the circles as red lines and duplication as blue lines. More intense is the color of the line, and higher is the heteroplasmy. The inner circle contains data related to the sequencing coverage. Higher is the thickness of the grey, higher is the number of read per position. Subfigure **B** and **C**: mapping of the mtDNA alterations among HEU. The cumulative number of deletions (**B**) or duplication (**C**) per position was plotted against a linear mitochondrial genome. Each line represents the three time points. The two origins of replication (O_H_ and O_L_) are located on top of the histogram as vertical arrows.

**Figure 2 biomedicines-10-01786-f002:**
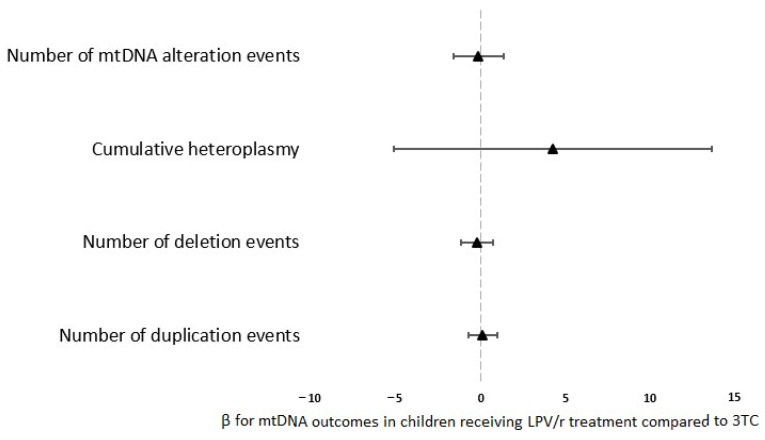
Forest plot of the association between LPV/r and 3TC prophylactic regimens and mtDNA alteration outcomes at W50. Regression coefficients (β) and their 95% confidence intervals are shown. Positive β values favor mtDNA outcomes at W50 in children receiving LPV/r treatment compared to those receiving 3TC treatment.

**Table 1 biomedicines-10-01786-t001:** Characteristics of HEU at the three time points.

	D7	W50	Y6
**Socio-Demographics**			
Site; *n (%)*			
Burkina Faso	13 (54.2)		
Zambia	11 (45.8)		
Sex; *n (%)*			
Boy	12 (50.0)		
Age (in weeks/years); *mean ± SD*	1.0 ± 0.0	1.0 ± 0.1 ^†^	6.3 ± 0.3
**Anthropometrics**; *mean ± SD*			
Weight (kg)	3.1 ± 0.4	8.1 ± 1.1 ^†^	19.1 ± 2.9
Height (cm)	49.0 ± 2.2	72.3 ± 3.0 ^†^	114.1 ± 5.2
Head circumference (cm)	N.A.	N.A.	50.8 ± 1.7 ^†^
WAZ	−0.7 ± 0.9	−1.2 ± 1.1 ^†^	−0.8 ± 1.0
HAZ	−0.9 ± 1.1	−0.9 ± 1.2 ^†^	−0.6 ± 1.0
BMIZ	−0.4 ± 1.5 ^†^	−0.9 ± 1.1 ^†^	−0.6 ± 0.9
Gestational age (week); *median [IQR]*	38.0 [37.0;39.5]	N.A.	N.A.
Preterm birth (week); *n (%)*			
No prematurity ≥ 37	21 (87.5)		
Prematurity < 37	3 (12.5)		
**General Examination**; *n (%)*			
Blood pressure, systolic (mmHg)			
Low < 95			12 (50.0)
Normal (95–100)	N.A.	N.A.	3 (12.5)
Elevated > 100			9 (37.5)
Elevated blood pressure, diastolic (mmHg)			
Normal (56–70)	N.A.	N.A.	14 (58.3)
Elavated > 70			10 (41.7)
Elevated pulse rate (rate/min)			
Normal (75–110)	N.A.	N.A.	23 (95.8)
Elevated > 110			1 (4.2)
Elevated respiratory rate (rate/min)			
Normal (21–23)	N.A.	N.A.	9 (37.5)
Elevated > 23			15 (62.5)
Oedema; *n (%)*		0 (0.0)	0 (0.0)
Clinical evaluation			
No abnormality			20 (83.3)
One abnormality	N.A.	N.A.	4 (16.7)
Blindness			1 (4.2)
Hearing impairment			1 (4.2)
Mild Arrhythmia			1 (4.2)
Heart murmur			1 (4.2)
**Hematology**; *n (%)*			
Hemoglobin concentration (g/dL); *n* (%)			
Normal	22 (91.7)	9 (52.9) ^§,^°	23 (95.8)
Anemia ^#^	2 (8.3)	8 (47.1) ^§,^°	1 (4.2)
Platelet count (10^3^/mm^3^); *n* (%)			
Normal > 125	24 (100.0)	17 (100.0) ^§,^°	24 (100.0)
Leucocyte count (10^3^/mm^3^); *n* (%)			
Normal > 2.5	24 (100.0)	17 (100.0) ^§,^°	24 (100.0)
Neutrophil count (10^3^/mm^3^); *n* (%)			
Normal	23 (100.0) ^†^	14 (82.4) ^§,^°	24 (100.0)
Neutropenia ^¥^	0.0 (0.0)	6 (17.7) ^§,^°	
**Biochemistry**; *n (%)*			
LDH concentration (xULN)	N.A.	N.A.	
Normal < ULN		N.A.	7 (29.2)
Abnormal ≥ ULN		N.A.	17 (70.8)
Grade 1 (Mild) [ULN-2xULN]		N.A.	12 (50.0)
Grade 2 (Moderate) [≥2xULN]		N.A.	5 (20.8)
ALT concentration (U/L)			
Normal < 1.25xULN	24 (100.0)	16 (94.1) ^§,^°	24 (100.0)
Abnormal ≥ 1.25xULN		1 (5.9) ^§,^°	
Grade 1 (Mild) [1.25–2.5]xULN		1 (5.9) ^§,^°	
**Medical events**; *n (%)*			
Clinical consultation without admission during the last year			
Yes	N.A.	N.A.	19 (79.2)
Hospital admission since the PROMISE-PEP trial			
Yes	N.A.	N.A.	5 (20.8)
Child ARV prophylaxis during the PROMISE-PEP trial			
Lopinavir/ritonavir (LPV/r)		12 (50.0)	12 (50.0)
**Development and neuropsychological assessment**			
10questionsPlus			
At least 1 disability; *n (%)*	N.A.	N.A.	7 (29.2)
SDQ-25; *mean ± SD*	N.A.	N.A.	7.7 ± 4.5
TOVA; *mean ± SD*	N.A.	N.A.	2.2 ± 0.7 ^†^
MABC-2; *mean ± SD*	N.A.	N.A.	75.6 ± 9.6 ^‡^
KABC-II; *mean ± SD*	N.A.	N.A.	57.4 ± 13.8 ^‡^

^†^ One missing value, ^‡^ two missing values, ^§^ seven missing values, ° data taken from the W38 visit as no laboratory tests were performed at the W50 visit, ^¥^ threshold for neutropenia was 1.5 at D7 and 1 at W50 and Y6, ^#^ threshold for anemia was 13 at D7 and 10.4 at W50 and Y6. Abbreviations: SD, standard deviation; NA, not applicable; WAZ, weight-for-age Z-score; HAZ, height-for-age Z-score; BMIZ, pediatric body mass index Z-score; LDH, Lactate dehydrogenase; ULN, Upper Limit of Normal; ALT, alanine transaminase; ARV, antiretroviral; SDQ-25, Strengths and Difficulties Questionnaire; TOVA, Test of Variable of Attention; MABC-2, Movement Assessment Battery for Children second edition; KABC-II, Kaufman Assessment Battery for Children second edition.

**Table 2 biomedicines-10-01786-t002:** Characteristics of the mother at D7 post-partum (*n* = 24).

Sociodemographic characteristics	
Age (year); *mean ± SD*	28.2 ± 5.6
Parity; *median [IQR]*	3.0 [1.5;3.0]
Education	
Mother/caregiver ever attended school; *n (%)*	
Yes	18 (75.0)
Clinical and biological characteristics	
BMI; *mean ± SD*	23.3 ± 2.8
CD4 cells count (cells/mm^3^); *median [IQR]*	543 [450;820]
HIV viral load control (copies/mL); *n (%)*	
<1000	14 (58.3)
≥1000	10 (41.7)
WHO HIV staging; *n (%)*	
Stage 1	24 (100.0)
Mode of delivery; *n (%)*	
Vaginal	24 (100.0)
Maternal prophylaxis during pregnancy	
ARV regimen; *n (%)*	
AZT	24 (100.0)
Duration of ARV prophylaxis (week); *median [IQR]*	8.0 [4.0;9.0]
Maternal lifestyle during pregnancy	
Smoking during pregnancy; *n (%)*	
No	22 (100.0) ^‡^
Alcohol consumption during pregnancy; *n (%)*	
Yes No	12 (54.6) ^‡^ 10 (45.5) ^‡^

^‡^ Two missing values. Abbreviations: D7, day seven; SD, standard deviation; IQR, interquartile range; BMI, body mass index; HIV, human immunodeficiency virus; WHO, World Health Organization; ARV, antiretroviral; AZT, azidothymidine or zidovudine.

**Table 3 biomedicines-10-01786-t003:** Characteristics of HEU with mtDNA alterations from D7 to Y6.

	D7 (*n* = 24)	W50 (*n* = 24)	Y6 (*n* = 24)
Child with mtDNA alteration; *n (%)*	16 (66.7)	20 (83.3)	18 (75.0)
Alteration events; *n*	36	50	33
Nb of event per affected child	2 [1.0–2.5]	2.0 [1.5–3.0]	1.0 [1.0–3.0]
Deletion	20	33	23
Class I/II/III	4/10/8	4/11/18	5/6/12
Duplication	16	17	10
Cumulative heteroplasmy (in %)			
Median [IQR]	0.24 [0.00–2.47]	0.43 [0.19–2.46]	0.24 [0.05–0.64]
Deletion heteroplasmy (in %)			
Median [IQR]	0.00 [0.00–0.48]	0.40 [0.06–1.38]	0.16 [0.00–0.36]
Duplication heteroplasmy (in %)			
Median [IQR]	0.03 [0.00–0.15]	0.00 [0.00–0.13]	0.00 [0.00–0.11]
Child with heteroplasmy > 1%; *n (%)*	7 (29.2)	9 (37.5)	3 (12.5)
Point mutation; *n*	0	0	0
**Child receiving 3TC PrEP (*n* = 12)**			
Child with mtDNA alteration; *n (%)*	9 (75.0)	10 (83.3)	8 (66.7)
Alteration events; *n*	22	26	12
Cumulative heteroplasmy			
Median [IQR]	0.41 [0.07–3.68]	0.32 [0.15–2.46]	0.13 [0.00–0.48]
Deletion class I/II/III	3/6/4	1/4/10	1/2/5
**Child receiving LPV/r PrEP (*n* = 12)**			
Child with mtDNA alteration; *n (%)*	7 (58.3)	10 (83.3)	10 (83.3)
Alteration events; *n*	14	24	20
Cumulative heteroplasmy			
Median [IQR]	0.08 [0.00–0.76]	0.54 [0.23–2.46]	0.35 [0.18–0.71]
Deletion class I/II/III	1/4/7	3/7/8	4/4/10

Abbreviations: mtDNA, mitochondrial DNA; IQR, interquartile range; 3TC, lamivudine; PrEP, Pre-exposure prophylaxis; LPV/r, liponavir/ritonavir.

**Table 4 biomedicines-10-01786-t004:** Determinants for having an elevated heteroplasmy rate at D7.

	N	Cumulative Hetereoplasmy Rate at D7		*p* Value
		*n (%)* > 1%	ρ	
Site				
Burkina Faso	13	5 (38.5)		0.36
Zambia	11	2 (18.2)		
Sex				
Boy	12	4 (33.3)		1.000
Girl	12	3 (25.0)		
Preterm birth				
No prematurity (≥37 weeks)	21	3 (33.3)		0.530
Premature (<37 weeks)	3	0 (0.0)		
Parity *(per child)*	24		0.034	0.876
Child weight at D7 *(per kg)*	24		−0.257	0.225
Mother’s age at birth				
≤30 years	14	4 (28.6)		0.643
30 years and over	10	3 (30.0)		
Mother education				
Yes	18	4 (22.2)		0.307
No	6	3 (50.0)		
Mother’s HIV viral load				
<1000 copies/mL	14	4 (28.6)		1.000
≥1000 copies/mL	10	3 (30.0)		
Duration of ARV prophylaxis during pregnancy *(per week)*	24		0.243	0.253
Alcohol consumption during pregnancy				
No	10	3 (30.0)		0.624
Yes	12	2 (16.7)		

Abbreviations: D7, day seven; ρ, rho (Spearman’s correlation coefficient); HIV, human immunodeficiency virus; ARV, antiretroviral.

**Table 5 biomedicines-10-01786-t005:** Association between having a heteroplasmy rate >1% at D7 and health outcomes at Y6.

Health Outcomes at Y6	Cumulative Heteroplasmy		*p*-Value
	rate > 1% at D7 *n* = 7	rate ≤ 1% at D7 *n* = 17	
**Growth** *mean* *±* *SD*			
Head circumference (cm)	50.7 ± 1.7	50.9 ± 1.7 ^†^	0.777
WAZ	−1.2 ± 0.7	−0.6 ± 1.1	0.142
HAZ	−0.9 ± 0.9	−0.4 ± 1.1	0.356
BMIZ	−1.0 ± 0.9	−0.4 ± 0.9	0.159
**Clinical evaluation**			
Systolic blood pressure Z-score; *mean* *±* *SD*	0.7 ± 1.0	0.3 ± 1.1	0.463
Clinical abnormalities; *n (%)*	1 (14.3)	3 (17.7)	1.000
Hospitalised since PROMISE PEP; *n (%)*	3 (42.9)	2 (11.8)	0.126
Abnormal LDH concentration; *n (%)*	5 (71.4)	12 (70.6)	1.000
Abnormal ALT concentration; *n (%)*	0 (0.0)	0 (0.0)	N.A.
**Developmental outcomes**			
At least one disability detected from the 10questionPlus; *n (%)*	3 (42.9)	4 (23.5)	0.374
SDQ-25; *mean* *±* *SD*	9.1 ± 4.4	7.1 ± 4.5	0.326
TOVA; *mean* *±* *SD*	2.1 ± 0.3	2.3 ± 0.8 ^†^	0.622
MABC-2; *mean* *±* *SD*	74.8 ± 9.0 ^†^	75.9 ± 10.1 ^†^	0.827
KABC-II; *mean* *±* *SD*	54.4 ± 21.5 ^†^	58.5 ± 10.4 ^†^	0.558

^†^ One missing value. Abbreviations: SD, standard deviation; WAZ, weight-for-age Z-score; HAZ, height-for-age Z-score; BMIZ, paediatric body mass index Z-score; LDH, Lactate dehydrogenase; ALT, alanine transaminase; SDQ-25, Strengths and Difficulties Questionnaire; TOVA, Test of Variable of Attention; MABC-2, Movement Assessment Battery for Children second edition; KABC-II, Kaufman Assessment Battery for Children second edition.

**Table 6 biomedicines-10-01786-t006:** Association between the duration of maternal prophylaxis during pregnancy and poor health outcomes at Y6, stratified by infant heteroplasmy rate (>1% or ≤1%) at D7.

Health Outcomes at Y6	β [95%CI] for Duration of Maternal Prophylaxis, Stratified by Infant Heteroplasmy Rate			*p*-Value for Interaction
	Total N = 24	>1% at D7 *n* = 7	≤1% at D7 *n* = 17	
**Growth**				
Head circumference (cm) ^†^	0.20 [−0.05–0.44]	0.31 [0.06–0.55]	−0.40 [−1.19–0.38]	0.033 *
WAZ	0.02 [−0.13–0.18]	−0.13 [−0.46–0.20]	0.06 [−0.10–0.23]	0.364
HAZ	0.06 [−0.10–0.22]	0.14 [−0.32–0.61]	0.05 [−0.12–0.23]	0.685
BMIZ	−0.02 [−0.17–0.12]	−0.34 [−0.66–−0.02]	0.04 [−0.10–0.19]	0.047 *
**Clinical evaluation**				
Systolic BP Z-score	0.09 [−0.06–0.24]	0.15 [−0.33–0.63]	0.08 [−0.10–0.25]	0.733
LDH concentration	−6.79 [−68.63–55.05]	20.91 [−65.57–107.39]	−7.55 [−76.26–61.16]	0.255
ALT concentration	0.40 [−0.76–1.56]	2.87 [−2.30–8.05]	−0.05 [−1.27–1.16]	0.070

^†^ One missing value, * statistically significant. Abbreviations: SD, standard deviation; WAZ, weight-for-age Z-score; HAZ, height-for-age Z-score; BMIZ, paediatric body mass index Z-score; BP, blood pressure; LDH, Lactate dehydrogenase; ALT, alanine transaminase.

## Data Availability

The data presented in this study are available in the Appendix A.

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
