# Peer review of "Long-Term Persistence of Mitochondrial DNA Instability in HIV-Exposed Uninfected Children during and after Exposure to Antiretroviral Drugs and HIV"

_biomedicines, 2022, doi:10.3390/biomedicines10081786_

Round 1

Reviewer 1 Report

The authors examine mtDNA instability in HEU infants in a longitudinal manner (to 6 years) and correlate with medical and developmental outcomes. Although the study is small, it uses a rather unique dataset with longitudinal data, so is novel. The reasons behind adverse outcomes in HEU infants remain unclear, and mitochondrial toxicity of ART has long been a hypothesis, but remains unproven. 

Specific comments.

Methods:

- Line 108 - D7 timepoint is described as 'before prophylaxis'. Is this correct, or was the infant prophylaxis started before D7 of life?

- Line 111 - How were the 24 cases selected? Presumably the associated PROMISE studies are very large, so what was the basis of this very small subgroup?

- Line 142-144 - three criteria are given for validation of an event. These are all presented as 'or', implying that any one of these could be present for the event to be valid. But should this not be 'and', that is all 3 criteria must be met for the event to be valid? Furthermore, a heteroplasmy level of 0.05% is extremely low. I am not aware of any data showing that events detected at such a low heteroplasmy level can be considered valid.

- Line 169 - cumulative heteroplasmy threshold of >1%. Why was this level chosen? Why is >5% also used elsewhere in the paper? Should be consistent, or systematically look at different levels of cumulative heteroplasmy.

Results:

- Line 208 - what determined whether child received 3TC or LPV/r? Was this randomised?

- Line 212 - the word 'child' is confusing as it is the mother that receives the AZT. Maternal treatment is described as 'AZT-containing', but is it actually the case that all mothers received AZT monotherapy only? Please clarify. If so, please also mention this is the discussion in the context of what ART mothers are likely to now be receiving during pregnancy.

- Line 215 - 'most HEU had abnormal cardiac parameters'. What does this mean? It implies perhaps abnormal ECG or TTE, which is not the case.

- Line 219-222 - there is text from instructions to authors that has not been deleted.

- Table 1 - provide information on mode of delivery

- Table 2 - move D7 data for infants into Table 2, leaving only maternal data in Table 1.

- Figure 1 - panel A legend describes one child at 3 timepoints, but there are 6 mito-circles. Is this actually 2 children at 3 timepoints? The final mito-circle appears to be rotated compared with the others, please correct. Are panels B and C each an individual child? Please clarify in the legend.

- Table 3 - please consider adding also number of events per child.

- Line 278 - please remove 'a trend towards statistical significance'. This is not supported by the data.

- Figure 2 - please add ART (favours 3TC, favours LPV/r) to x-axis.

- Line 295 - suggest 'presence of' rather than 'being diagnosed with' (this is not a diagnostic test). Clarify that 'heteroplasmy rate' here means cumulative.

- Line 312-337 and Table 6 - I do not agree with the approach to stratify by infant heteroplasmy rate and perform interaction testing. The numbers are simply too small for this. Suggest remove this section.

Discussion:

- Line 373 - clarify choice of children for mitokine analysis and expand on score used and cut-off (and move this to methods not discussion). 

- Line 379-383 - I find the comparison with urothelial cells to be very indirect and of doubtful relevance. It is also too speculative to talk about premature aging of blood cells, especially when inferring this from published data on urothelial cells. If the authors wish to hyopthesise premature aging of blood cells they need to cite comparable data from blood to support this.

- Line 393-394 - the authors mention two specific rearrangements. Are these described elsewhere in the literature?

- Line 401-403 - the authors mention exposure to maternal HIV proteins. Is it known how much maternal HIV protein a baby is actually exposed to - in utero and through breastmilk? Is it not more plausible that the altered maternal physiology / metabolism (compared to an HIV negative mother) is more relevant?

- Line 413 - I find the comparison of lifespan with Patas monkey to be spurious as it is likely to be the rate of turnover of leucocytes and/or of mtDNA that is relevant here and this may be little different between the Patas monkey and the human.

- Addition - there needs to be further explanation of why the mtDNA abnormalities seem to persist at Y6, so long after the cessation of ART. The authors do not offer any very plausible explanation for this. I would suggest that this might suggest that blood stem cells have sustained mtDNA damage. 

- Line 421 - were the authors surprised that types of deletion did not differ with 3TC vs. LPV/r exposure? Might these be expected to have different mechanisms of mitochondrial toxicity? Alternatively, I would suggest that this may suggest that it is not the infant ART exposure that is causing these mtDNA abnormalities, but rather a persistent effect of the maternal AZT or other maternal factors. Again, this could well have affected the blood stem cells.

- Line 431 - suggest 'could be' rather than 'should be considered'.

- Line 443 - suggest 'archive' rather than 'backlog'.

- Line 457 - comment re. premature aging is too speculative.

Supplementary tables:

- I would like to see a table listing each rearrangements detected in each case. It is very hard to make sense of the cumulative heteroplasmy levels without this.

Author Response

Answers to Reviewer#1. Lines are given from the revised manuscript without tracking

Comments and Suggestions for Authors

The authors examine mtDNA instability in HEU infants in a longitudinal manner (to 6 years) and correlate with medical and developmental outcomes. Although the study is small, it uses a rather unique dataset with longitudinal data, so is novel. The reasons behind adverse outcomes in HEU infants remain unclear, and mitochondrial toxicity of ART has long been a hypothesis, but remains unproven. 

Specific comments.

Methods:

- Line 108 - D7 timepoint is described as 'before prophylaxis'. Is this correct, or was the infant prophylaxis started before D7 of life?

We agree that the description was partly correct. All of these children received 6 days of nevirapine from birth, as per WHO recommendation at the time of the study. The text was modified as follows: Line 104: “long-term effects of extended ARV prophylaxis” and line 110: “and received six days of nevirapine from birth followed by one year of lamivudine or lopinavir/ritonavir extended prophylaxis”. 

- Line 111 - How were the 24 cases selected? Presumably the associated PROMISE studies are very large, so what was the basis of this very small subgroup?

The selection was done sequentially during the PhD work of Audrey Monnin. We added in a supplementary figure of the sample flow selection.

- Line 142-144 - three criteria are given for validation of an event. These are all presented as 'or', implying that any one of these could be present for the event to be valid. But should this not be 'and', that is all 3 criteria must be met for the event to be valid? Furthermore, a heteroplasmy level of 0.05% is extremely low. I am not aware of any data showing that events detected at such a low heteroplasmy level can be considered valid.

Thank you for pointing out this error. Indeed, all three criteria need to be met for validation. We agree with the reviewer that the threshold is low, at least from a clinical point of view. However, assays were performed on DNA extracted from blood leukocytes which is not the tissue with the most prevalent mtDNA instability, and infraclinic events are worth reporting as they may amplify with time. This sensitivity in detecting low levels of heteroplasmy allows to revisit the issue of mtDNA instability (cumulative effect of many variants with low levels of heteroplasmy) like in various other studies (Ultrasensitive deletion detection links mitochondrial DNA replication, disease, and aging, Genome Biol. 2020 Sep 17;21(1):248. doi: 10.1186/s13059-020-02138-5; Mitochondrial DNA content reduction in the most fertile spermatozoa is accompanied by increased mitochondrial DNA rearrangement Hum Reprod. 2022 Apr 1;37(4):669-679.doi: 10.1093/humrep/deac024; Heteroplasmy of mutant mitochondrial DNA A10398G and analysis of its prognostic value in non-small cell lung cancer. Oncol Lett. 2016 Nov;12(5):3081-3088.doi: 10.3892/ol.2016.5086.) including one at the single cell level (Identification of unique and shared mitochondrial DNA mutations in neurodegeneration and cancer by single-cell mitochondrial DNA structural variation sequencing (MitoSV-seq) EBioMedicine. 2020 Jul;57:102868.doi: 10.1016/j.ebiom.2020.102868).

- Line 169 - cumulative heteroplasmy threshold of >1%. Why was this level chosen? Why is >5% also used elsewhere in the paper? Should be consistent, or systematically look at different levels of cumulative heteroplasmy.

To date, a significant level for this threshold has not been set by the scientific community. Analysis could have been done by using heteroplasmy as a continuous variable. However, given the low number of cases, a categorical approach was chosen. A threshold of 1% is not a clinical warning sign but can still be used to clearly illustrate a strong genetic instability, which is the main message of this manuscript. We removed all results related to a threshold of 5% which was also tested but not more informative.

- Line 208 - what determined whether child received 3TC or LPV/r? Was this randomised?

Yes, the participants were randomised in PROMISE-PEP trial. The text was modified accordingly. Line 111, “Children were randomly assigned to one of the two arms at day 7 as per PROMISE-PEP protocol (NCT00640263).

- Line 212 - the word 'child' is confusing as it is the mother that receives the AZT. Maternal treatment is described as 'AZT-containing', but is it actually the case that all mothers received AZT monotherapy only? Please clarify. If so, please also mention this is the discussion in the context of what ART mothers are likely to now be receiving during pregnancy.

At the time of enrolment in PROMISE-PEP, mothers were not eligible to receive ARV treatment as per WHO recommendations. The inclusion criteria for the trial included mothers with CD4 cell counts above 350/mL. Notwithstanding, mothers received AZT monotherapy during pregnancy to prevent HIV transmission to their child. We added to the text: line 206, “Mothers were not eligible to receive ART treatment at the time of the trial as per WHO’s recommendations but all received an antenatal AZT-containing regimen for a median duration of eight weeks to prevent perinatal HIV transmission.

- Line 215 - 'most HEU had abnormal cardiac parameters'. What does this mean? It implies perhaps abnormal ECG or TTE, which is not the case.

We corrected the sentence by “most HEU had abnormal blood pressure”.

- Line 219-222 - there is text from instructions to authors that has not been deleted.

Thank you for pointing this out. The text was deleted.

- Table 1 - provide information on mode of delivery

This information has now been added to Table 2.

- Table 2 - move D7 data for infants into Table 2, leaving only maternal data in Table 1.

The tables were modified accordingly.

- Figure 1 - panel A legend describes one child at 3 timepoints, but there are 6 mito-circles. Is this actually 2 children at 3 timepoints? The final mito-circle appears to be rotated compared with the others, please correct. Are panels B and C each an individual child? Please clarify in the legend.

We corrected and modified the figure as well was the figure caption to answer reviewer’s comments.

- Table 3 - please consider adding also number of events per child.

The data is now in the table.

- Line 278 - please remove 'a trend towards statistical significance'. This is not supported by the data.

The text was modified as follows: Line 280 None of the tested determinants were statistically associated

- Figure 2 - please add ART (favours 3TC, favours LPV/r) to x-axis.

The figure’s x axis has now been labelled to allow readers to more clearly identify that the figure compares ART regimens and that 3TC is the reference group for the presented beta values.

- Line 295 - suggest 'presence of' rather than 'being diagnosed with' (this is not a diagnostic test). Clarify that 'heteroplasmy rate' here means cumulative.

We modified the text as follows: Detection of a cumulative

- Line 312-337 and Table 6 - I do not agree with the approach to stratify by infant heteroplasmy rate and perform interaction testing. The numbers are simply too small for this. Suggest remove this section.

We thank the reviewer for their suggestion and fully agree that the numbers are too small for this analysis to provide any conclusive results regarding the role an elevated heteroplasmy plays in HEU growth and development. However, we think that the added value of this stratification is important as it continues our exploratory analysis and provides an interesting insight into the possible nature of the relationship between these parameters. We already checked the language of the text to emphasize the fact that these results are in no way conclusive but simply deserving of further investigation using a larger sample and a control group, as suggested by a number of other comments by the reviewers.

Discussion:

- Line 373 - clarify choice of children for mitokine analysis and expand on score used and cut-off (and move this to methods not discussion). 

We decided to remove the mitokine substudy from the manuscript because the design of the experiments was not optimal. As pointed out by other reviewers, time-points for mtDNA alterations and mitokine detection were too different. While mtDNA alteration was done using DNA-extracts from DBS, we did not have in the sample repository matched plasma or serum. Further developments on mitokine dosages from DBS eluates are needed.

- Line 379-383 - I find the comparison with urothelial cells to be very indirect and of doubtful relevance. It is also too speculative to talk about premature aging of blood cells, especially when inferring this from published data on urothelial cells. If the authors wish to hyopthesise premature aging of blood cells they need to cite comparable data from blood to support this.

We agree with the reviewer that the conclusion of premature aging of blood cells is too speculative. The text was modified: we replace premature aging by early mtDNA instability. However, our purpose was to position the observed values with those of the literature. Unfortunately, NGS applied to blood cells for the detection of mtDNA deletion cannot be found in the literature. Uroepithelial cells are commonly used in our diagnostic activities for mitochondrial diseases because they usually express higher levels of heteroplasmy of pathogenic variants. The comparison with DNA from these cells in our paper therefore seems to us an interesting control that confirms the accuracy of our results obtained on blood DNA. We added to the text: Line 368 Bris et al. reported using the same computational pipeline an average of 0.18% deletion rate in uroepithelial cells of healthy persons aged 50 years and more[11]. Uroepithelial cells are commonly used in our diagnostic activities for mitochondrial diseases because they usually express higher levels of heteroplasmy of pathogenic variants. Our results showed levels of mtDNA rearrangements in blood HEU similar to those found in urine from a population over 50 years old, suggesting early mtDNA instability in blood cells.

- Line 393-394 - the authors mention two specific rearrangements. Are these described elsewhere in the literature?

We did not find any previous descriptions of these rearrangements in the literature or in databases such as Mitobreak http://mitobreak.portugene.com/.

- Line 401-403 - the authors mention exposure to maternal HIV proteins. Is it known how much maternal HIV protein a baby is actually exposed to - in utero and through breastmilk? Is it not more plausible that the altered maternal physiology / metabolism (compared to an HIV negative mother) is more relevant?

We agree with the reviewer, others factors related to the mother HIV infection can participate to the risk factors. We added to the text Line 397: While ARV exposure can be characterized as type and dose of ARV and duration of ex-posure, HIV exposure is more complex and more difficult to estimate. Such exposure in-cludes items such as exposure to free HIV proteins already known as mitotoxicant(43), but also the known microinflammation of all HIV-infected patients or their altered metabo-lism. Deciphering between those three and others is not technically possible, yet.

- Line 413 - I find the comparison of lifespan with Patas monkey to be spurious as it is likely to be the rate of turnover of leucocytes and/or of mtDNA that is relevant here and this may be little different between the Patas monkey and the human.

The comparison was removed from the text. We modified the text as followed: “Noticeably, at three years of age, mitochondrial toxicity was still detectable, suggesting that . Given that three years of age for Patas monkey approximately corresponds to 15 years of age for a human, the rate of elimination after exposure discontinuation appearswas slow[43,44].

- Addition - there needs to be further explanation of why the mtDNA abnormalities seem to persist at Y6, so long after the cessation of ART. The authors do not offer any very plausible explanation for this. I would suggest that this might suggest that blood stem cells have sustained mtDNA damage. 

We added the following text Line 378: Furthermore, the long-lasting detection of such instability, up to six years of age, together with the rapid renewal of blood cells and the discontinuation of the HIV and/or ARV ex-posure suggest that these alterations occur also in blood stem cells. It also suggests that the replicative advantage of the deleted molecules outweighed their physiological elimination processes.

- Line 421 - were the authors surprised that types of deletion did not differ with 3TC vs. LPV/r exposure? Might these be expected to have different mechanisms of mitochondrial toxicity? Alternatively, I would suggest that this may suggest that it is not the infant ART exposure that is causing these mtDNA abnormalities, but rather a persistent effect of the maternal AZT or other maternal factors. Again, this could well have affected the blood stem cells.

We agree with the reviewer and added to the text: Line 429 It is noteworthy that all of the HEU of this short case series were also exposed to maternal AZT treatment during pregnancy. Therefore, we cannot definitively rule out that the observed mtDNA instability in HEU results from the in utero maternal AZT treatment, similarly to what was reported in Patas monkey experiments. Analysing a group of mothers not receiving treatment is needed.

- Line 431 - suggest 'could be' rather than 'should be considered'.

We have modified the text accordingly.

- Line 443 - suggest 'archive' rather than 'backlog'.

We have modified the text accordingly.

- Line 457 - comment re. premature aging is too speculative.

We agree with the reviewer, aging is a complex process involving more than mtDNA instability. We replace premature aging by early mtDNA instability.

Supplementary tables:

- I would like to see a table listing each rearrangements detected in each case. It is very hard to make sense of the cumulative heteroplasmy levels without this.

We provided the list of all the events that passed the cut-off of validation as supplementary file.

Thank you for helping to improve the manuscript.

Reviewer 2 Report

The manuscript by Desquiret-Dumas et al describes an interesting and understudied phenomenon of mitochondrial DNA instability in HEU children after ART and HIV exposure. The manuscript is well written and most of the methods are well described. However, there are some aspects of the manuscript that require further detail or clarification.

While it is stated later in the manuscript that this is an exploratory study, a statement on why such a small sample size was used could be included in the abstract. Were only 24 samples available that met the criteria? 

In section 2.2, the clade of HIV virus should be included.  Was this clade C?  Most virus in the United States and Europe is clade B. 

In section 2.5, you stated that the heteroplasmy at D7 was associated with W6 mitokines, and W38 heteroplasmy with W50 mitokines.  Some discussion should be offered as to why the time frames between heteroplasmy determination and myokine measurements were disparate, and why they were so far apart from each other? If the heteroplasmy in peripheral blood cells is transient, and the heteroplasmy is offered as a stimulator of mitokine expression, I would expect the causal relationship would require measurements in closer temporal proximity.  Are samples available to measure the mitokines at a set time after heteroplasmy measures?  I would think you might want to measure mitokines 1 week after heteroplasmy measures, for example.  Ideally, a time course of mitokine measures would be performed.

The last 2 sentences of section 3.1 seem to be instructions for the writing the results section.

Is there any further data available on alcohol consumption during pregnancy?  It would be helpful to know quantity and frequency.

Figure 1A is very low resolution.  Please use a higher resolution figure.  I would suggest adding labels to figure 1A for the time points, and for what the two rows are.  Overall, a better description of Figure 1A should be given. I would prefer not to have to go to reference 11 for the details if they could be summarized succinctly here. Does each row represent a different child?  Consider using different dashed lines for the different time points in figures 1B and 1C.  Also, it is not clear where the origins of replication are located as stated in the figure legend.

The description of the data from lines 253 to 258 was confusing.  Either the values in the text do not match the table or I am misreading something.  Similarly, lines 262-264 show the correct percentages, but the IQR doesn’t match.  Please clarify.

Line 334: Should “ended” be “tended”?

On lines 412-414, discussion of Patas monkey and human age equivalency should be removed. These equivalence estimations are important in specific fields of study, such as development. It’s highly doubtful that the mitochondrial toxicity would last 15 years in a human.

Author Response

Answers to Reviewer#2. Lines are given from the revised manuscript without tracking

The manuscript by Desquiret-Dumas et al describes an interesting and understudied phenomenon of mitochondrial DNA instability in HEU children after ART and HIV exposure. The manuscript is well written and most of the methods are well described. However, there are some aspects of the manuscript that require further detail or clarification.

While it is stated later in the manuscript that this is an exploratory study, a statement on why such a small sample size was used could be included in the abstract. Were only 24 samples available that met the criteria? 

We agree with the reviewer and have modified the text accordingly. Line 25: “We aimed to explore the relationship between these health outcomes and mtDNA deletions overtime in a case series of 24 HEU children.

In section 2.2, the clade of HIV virus should be included.  Was this clade C?  Most virus in the United States and Europe is clade B. 

We have added this information in the section 2.2 as follows: Line 112: “In Zambia, circulating HIV viruses were of clade C and in Burkina Faso of subtype CRF2 and CRF06.”

In section 2.5, you stated that the heteroplasmy at D7 was associated with W6 mitokines, and W38 heteroplasmy with W50 mitokines.  Some discussion should be offered as to why the time frames between heteroplasmy determination and myokine measurements were disparate, and why they were so far apart from each other? If the heteroplasmy in peripheral blood cells is transient, and the heteroplasmy is offered as a stimulator of mitokine expression, I would expect the causal relationship would require measurements in closer temporal proximity.  Are samples available to measure the mitokines at a set time after heteroplasmy measures?  I would think you might want to measure mitokines 1 week after heteroplasmy measures, for example.  Ideally, a time course of mitokine measures would be performed.

MtDNA analysis was done from DNA extracted from DBS. We did not have time point-matched serum or plasma in the sample repository to perform the mitokine analysis. We fully agree that the design of this substudy is questionable. We decided to remove these results from the revised manuscript.

The last 2 sentences of section 3.1 seem to be instructions for the writing the results section.

We thank the reviewer for pointing this out to us, the text has been revised accordingly.

Is there any further data available on alcohol consumption during pregnancy?  It would be helpful to know quantity and frequency.

The case report forms from PROMISE PEP did not collect any data concerning alcohol consumption during pregnancy. This information came from the PROMISE M&S trial and is self-reported

Figure 1A is very low resolution.  Please use a higher resolution figure.  I would suggest adding labels to figure 1A for the time points, and for what the two rows are.  Overall, a better description of Figure 1A should be given. I would prefer not to have to go to reference 11 for the details if they could be summarized succinctly here. Does each row represent a different child?  Consider using different dashed lines for the different time points in figures 1B and 1C.  Also, it is not clear where the origins of replication are located as stated in the figure legend.

We apologize that we did not check the quality of the figure after the pdf transformation. The different comments served to improve the figure and the figure caption was revised.

The description of the data from lines 253 to 258 was confusing.  Either the values in the text do not match the table or I am misreading something.  Similarly, lines 262-264 show the correct percentages, but the IQR doesn’t match.  Please clarify.

For the data in lines 242 to 247, these values are correct but are not presented in the table. The 55% at D7 and 69% at Y6 are the result of 20 deletion events out of the total 36 mtDNA alteration events at D7 (20/36=0.55) and 23 deletion events out of the total 33 mtDNA alteration events at Y6 (23/33=0.69). The rest of the data are neither shown nor drawn from any data presented in the table. For the data in lines 251-253 we have now specified that these values are in fact not the IQR but the minimum to maximum value range for heteroplasmy at each time point.

Line 334: Should “ended” be “tended”?

This is now corrected

On lines 412-414, discussion of Patas monkey and human age equivalency should be removed. These equivalence estimations are important in specific fields of study, such as development. It’s highly doubtful that the mitochondrial toxicity would last 15 years in a human.

We have modified the text as follows: “Noticeably, at three years of age, mitochondrial toxicity was still detectable, suggesting that . Given that three years of age for Patas monkey approximately corresponds to 15 years of age for a human, the rate of elimination after exposure discontinuation appearswas slow[43,44].

Thank you for helping to improve the manuscript.

Reviewer 3 Report

This is a secondary use of data study of 24 children HIV-exposed but uninfected who had been participants of the one of two studies and received 3TC or LPV/r prophylaxis during the first year of life, starting on day 7. Blood specimens collected on day 7, week 5 and year 6 were assayed for mtDNA deletion using the eKlipse-v2 methodology. The authors analyze several outcome measures related to mtDNA stability and attempt to relate those to as many as 20+ variables related to growth and clinical outcomes later in life. Though mtDNA alterations are observed, few associations are found.

Generally, this is an interesting topic that needs to be studied. However, there are important flaws and concerns as described below. The present study a highly exploratory and inadequately powered to assess these associations which greatly reduces its relevance apart perhaps to generate hypotheses guiding future studies. As can be expected from underpowered studies, very few associations are observed but of course in this case, the absence of evidence is not evidence of absence.

Major comments:

1.       The most important flaw is that although this is a repeated measure study, the analyses are carried out as if specimens collected at different times over a child’s life were independent, which of course they are not. The statistics should employ paired tests, or repeated measure tests.

2.       The objectives of the study are well described and laudable. However, as stated above, the study is grossly underpowered to “identify the associated risk factors and potential association with health outcomes later in life”. Everywhere in this manuscript, the word “explore” should be used in lieu of “assessed” (e.g. line 29,) “document” (e.g. line 100), determine, etc,.and the word “suggest” should be used in describing the findings and their potential significance.

3.       Line 455. The term unexpectedly should be stricken as this study does not include a control group that would allow onto determine whether this is an expected result or not. Reference #28 that describes 75% mtDNA alteration in CHEU vs. 0% in control CHUU is deeply flawed as the controls were in a different country, from a different population, with specimens processed and stored differently, at a different time, etc. Until a properly designed study shows how mtDNA alterations evolve over time during childhood, this type of conclusion is overreaching.

4.       Lines 89-91. Misleading and erroneous citing of the literature. The references provided do not support the following statement “…  one can say that HEU have an intermediate mtDNA copy number compared to HUU and HIV-infected children at birth, which would then return to a normal value upon discontinuation of ARV drugs exposure[19–21] “. Reference #19 is a very small study (n<20/group) reporting decreased mtDNA content in HEU vs HUU, reference 20 is a much larger study (n=214/group) reporting increased mtDNA content in HEU vs HUU that persists well past the discontinuation of ARV exposure, and reference #21 is a moderate size study (n=38 to 75 /group) reporting longitudinal mtDNA depletion in HEU+ ARV prophylaxis over one year, but still increased mtDNA vs. HEU who did not receive prophylaxis, with results that seem to vary greatly according to study site which is rather strange given that the children received similar HIV/ARV exposure…

It is a disservice to the readers to make them believe there is consensus in the literature over this issue since clearly there is not. Further, there seems to be concerning cherry picking of the literature cited where for example, findings by McComsey et al (2008) and Ross et al. (2012) are not mentioned but also contradict what the authors casually state in their introduction, as if this was a well-accepted state of knowledge.

5.       Line 98. The word “significant” should only be used if a significant p value was reported. Otherwise, the word “substantial” would be preferable.

6.        

7.       Line 111. 24 CHEU were selected…How was that selection done? Was it random? Was it all the children with complete study specimens? If yes, this should be stated. If not, the authors must explain how the selection was done as this could introduce bias in the study. For example, if these were selected based on other mtDNA data, this should be transparently reported. As per STROBE guidelines, a schematic of all trial participants and the criteria applied to select the final study sample should be included in supplement.

8.       This reviewer does not understand the rationale for and value of assaying mitokines on a seemingly random set of 11 plasma samples collected at different times (W6 and W38) than the one used to study mtDNA instability (D7, W50, and Y6). And then associate D7 with W6 and W38 with W50. Although it is clear that this was done for convenience, no rationale is presented to validate this approach.

9.       Line 169-171. A total of 9 potential risk factor “…study site, sex, gestational age, parity, mother’s age at birth, mother education, mother HIV viral load at D7, duration of ARV prophylaxis during pregnancy, and alcohol consumption during pregnancy” and 14 indicators of child’s health at 6 years of age were investigated for their association with heteroplasmy at D7, for a total of 23 tests of association among a total of 24 children, without any correction for multiple comparisons. The risk of a type 1 error is extremely high given that 1/20 test will yield a significant p value, more if the factors are not independent from one another which clearly is the case here.

10.   Line 202. The authors acknowledge the exploratory nature of the models presented and the fact that these are based on only 24 HEU. However, this statement is completely hidden in the methods section and should instead be made in the abstract and in the conclusion paragraph of the manuscript. They also acknowledge the limitations of the study but do not use language that is in line with those limitations. In general, the language needs to be toned down to reflect the high degree of uncertainty.

11.   Line 33-35. Given that the study was small and underpowered, the language should be edited to: In this small study sample, we did not detect any association between any health outcome at 6 years and mtDNA instability measures. Just because you did not detect it does not mean it does not exist and the language should reflect this. Absence of evidence is not evidence of absence, especially in underpowered studies.

12.   Line 38. Given the exploratory nature of the study and the lack of a control group, the language should be toned down. Genetic instability appears to persist over time among HEU yet remains sub-clinical.

13.   The text and tables report using year 6 specimens but figure 1 reports year 7. This should be clarified.

14.   The authors have access to longitudinal samples from 24 participants. Their analyses must therefore link samples from a given participants and not report data as if these were three independent groups. Repeated measures or paired statistics must be used as opposed to simple student t-tests.

15.   Transparency in data reporting. In Table 3, the authors must reveal whether it is the same children who have heteroplasmy at each time point or if these are random as this would substantially alter how the reader will interpret these results.

16.   Figure 2 needs a label for the X axis and the fact that CHEU exposed to 3TC are the reference group should be made clearer in the first line of the figure caption.

17.   Given the very large number of associations explored, the likelihood of a type 1 error is substantial yet no correction for multiple comparison were employed. In the section on study limitations, the authors should state that given the exploratory nature of the study, no correction for multiple comparison was applied.

Minor comments

1.       The maternal PBMCs were washed in PBS containing 2% FBS. It is highly unusual to risk contamination of human mtDNA with bovine mtDNA this way. Did the authors ascertain that the primers used to amplify the mtDNA did not also amplify bovine mtDNA?

2.       Line 141. The mean ± SD number of reads…

3.       Presumably children under the age of 7 have not yet decided on their gender identity. Sex not gender should be used throughout.

4.       There should be consistency on the number of decimals used in Tables (e.g. 58.5 ± 10.4, not 10.41)

Author Response

Answers to Reviewer#3. Lines are given from the revised manuscript without tracking

This is a secondary use of data study of 24 children HIV-exposed but uninfected who had been participants of the one of two studies and received 3TC or LPV/r prophylaxis during the first year of life, starting on day 7. Blood specimens collected on day 7, week 5 and year 6 were assayed for mtDNA deletion using the eKlipse-v2 methodology. The authors analyze several outcome measures related to mtDNA stability and attempt to relate those to as many as 20+ variables related to growth and clinical outcomes later in life. Though mtDNA alterations are observed, few associations are found.

Generally, this is an interesting topic that needs to be studied. However, there are important flaws and concerns as described below. The present study a highly exploratory and inadequately powered to assess these associations which greatly reduces its relevance apart perhaps to generate hypotheses guiding future studies. As can be expected from underpowered studies, very few associations are observed but of course in this case, the absence of evidence is not evidence of absence.

Major comments:

  1. The most important flaw is that although this is a repeated measure study, the analyses are carried out as if specimens collected at different times over a child’s life were independent, which of course they are not. The statistics should employ paired tests, or repeated measure tests.

We will respond in detail to this comment in point number 14 that covers the same issue.

  1. The objectives of the study are well described and laudable. However, as stated above, the study is grossly underpowered to “identify the associated risk factors and potential association with health outcomes later in life”. Everywhere in this manuscript, the word “explore” should be used in lieu of “assessed” (e.g. line 29,) “document” (e.g. line 100), determine, etc,.and the word “suggest” should be used in describing the findings and their potential significance.

We agree with the reviewer and have gone through the text to employ most appropriate words.

  1. Line 455. The term unexpectedly should be stricken as this study does not include a control group that would allow onto determine whether this is an expected result or not. Reference #28 that describes 75% mtDNA alteration in CHEU vs. 0% in control CHUU is deeply flawed as the controls were in a different country, from a different population, with specimens processed and stored differently, at a different time, etc. Until a properly designed study shows how mtDNA alterations evolve over time during childhood, this type of conclusion is overreaching.

We remove the term “unexpectedly” from the conclusion section.

  1. Lines 89-91. Misleading and erroneous citing of the literature. The references provided do not support the following statement “… one can say that HEU have an intermediate mtDNA copy number compared to HUU and HIV-infected children at birth, which would then return to a normal value upon discontinuation of ARV drugs exposure[19–21] “. Reference #19 is a very small study (n<20/group) reporting decreased mtDNA content in HEU vs HUU, reference 20 is a much larger study (n=214/group) reporting increased mtDNA content in HEU vs HUU that persists well past the discontinuation of ARV exposure, and reference #21 is a moderate size study (n=38 to 75 /group) reporting longitudinal mtDNA depletion in HEU+ ARV prophylaxis over one year, but still increased mtDNA vs. HEU who did not receive prophylaxis, with results that seem to vary greatly according to study site which is rather strange given that the children received similar HIV/ARV exposure…

It is a disservice to the readers to make them believe there is consensus in the literature over this issue since clearly there is not. Further, there seems to be concerning cherry picking of the literature cited where for example, findings by McComsey et al (2008) and Ross et al. (2012) are not mentioned but also contradict what the authors casually state in their introduction, as if this was a well-accepted state of knowledge.

We agree with the reviewer and we have removed the sentence.

  1. Line 98. The word “significant” should only be used if a significant p value was reported. Otherwise, the word “substantial” would be preferable.

The text was modified accordingly.

  1. Line 111. 24 CHEU were selected…How was that selection done? Was it random? Was it all the children with complete study specimens? If yes, this should be stated. If not, the authors must explain how the selection was done as this could introduce bias in the study. For example, if these were selected based on other mtDNA data, this should be transparently reported. As per STROBE guidelines, a schematic of all trial participants and the criteria applied to select the final study sample should be included in supplement.

The selection was done sequentially during the PhD work of Audrey Monnin. We added in supplementary figure the sample flow selection. A comparison of the randomly selected participants that were not included in this analysis (n=87 from Zambia and Burkina Faso only) versus the 24 that were included was performed. There were no statistically significant differences in baseline characteristics, expect that included mothers had more often consumed alcohol during their pregnancy compared to those that were not included (p=0.013). And although not statistically significant, the 24 mothers included in the sub study appeared to have taken AZT containing prophylaxis for a shorter amount of time (p=0.088). The comparison is now shown in Supplementary Table S1.

  1. This reviewer does not understand the rationale for and value of assaying mitokines on a seemingly random set of 11 plasma samples collected at different times (W6 and W38) than the one used to study mtDNA instability (D7, W50, and Y6). And then associate D7 with W6 and W38 with W50. Although it is clear that this was done for convenience, no rationale is presented to validate this approach.

We have removed this substudy from the manuscript. We agree that the design was set with the constraint of sample availability and does not to optimally answer the question. We will test this association with a more appropriate set of samples in the near future.

  1. Line 169-171. A total of 9 potential risk factor “…study site, sex, gestational age, parity, mother’s age at birth, mother education, mother HIV viral load at D7, duration of ARV prophylaxis during pregnancy, and alcohol consumption during pregnancy” and 14 indicators of child’s health at 6 years of age were investigated for their association with heteroplasmy at D7, for a total of 23 tests of association among a total of 24 children, without any correction for multiple comparisons. The risk of a type 1 error is extremely high given that 1/20 test will yield a significant p value, more if the factors are not independent from one another which clearly is the case here. $

We agree with the reviewer and this was noted in Line 197 as follows: “None of the models were adjusted for potentially confounding factors as this analysis is exploratory and based on a small sample of only 24 HEU.” Upon your comments, we have repeated, in the limitation section, that this approach is one of the limitations of the manuscript.

  1. Line 202. The authors acknowledge the exploratory nature of the models presented and the fact that these are based on only 24 HEU. However, this statement is completely hidden in the methods section and should instead be made in the abstract and in the conclusion paragraph of the manuscript. They also acknowledge the limitations of the study but do not use language that is in line with those limitations. In general, the language needs to be toned down to reflect the high degree of uncertainty.

The text has been modified accordingly.

  1. Line 33-35. Given that the study was small and underpowered, the language should be edited to: In this small study sample, we did not detect any association between any health outcome at 6 years and mtDNA instability measures. Just because you did not detect it does not mean it does not exist and the language should reflect this. Absence of evidence is not evidence of absence, especially in underpowered studies.

The text has been modified accordingly.

  1. Line 38. Given the exploratory nature of the study and the lack of a control group, the language should be toned down. Genetic instability appears to persist over time among HEU yet remains sub-clinical.

We agree with the reviewer and the language of the manuscript has been revised to better portray the exploratory nature of the study and its’ findings.

  1. The text and tables report using year 6 specimens but figure 1 reports year 7. This should be clarified.

Thank you for pointing out this error. It has now been corrected.

  1. The authors have access to longitudinal samples from 24 participants. Their analyses must therefore link samples from a given participants and not report data as if these were three independent groups. Repeated measures or paired statistics must be used as opposed to simple student t-tests.

We thank the reviewer for this suggestion. However, the statistical tests performed do not concern any longitudinal analyses. Student t-tests were only performed for the determinants of an elevated heteroplasmy rate at D7 (Table 4) which only includes baseline data in both dependent and explanatory variables, and for association between poor health outcomes at Y6 and elevated heteroplasmy at D7 (Table 5)—in this analysis we are not as interested in studying the evolution of these outcomes in relation to the evolution of heteroplasmy, but more so the possible use of an elevated heteroplasmy rate at D7 as an indicator for poor health outcomes down the line, and thus we believe that a simple student t-test is sufficient for the exploratory risk factor analyses we conducted. The forest plot coefficients and their 95% confidence intervals have been modified to follow the reviewer’s suggestion, as opposed to a classic linear regression model, repeated measures were taken into account using a mixed model. The methods section has been modified accordingly.

  1. Transparency in data reporting. In Table 3, the authors must reveal whether it is the same children who have heteroplasmy at each time point or if these are random as this would substantially alter how the reader will interpret these results.

For transparency, we added as a supplementary file the list of the 119 alteration events, with their position and heteroplasmy. Adding a line in the table 3 that summarise the different profiles looks tricky. We added to the text line 238: “All but one of the children with alteration events at D7 had alterations at W50. Few children acquired alteration events at W50 (n= 7) and at Y6 (n=1) (see supplementary table 1).”

  1. Figure 2 needs a label for the X axis and the fact that CHEU exposed to 3TC are the reference group should be made clearer in the first line of the figure caption.

The figure has been modified for easier reading.

  1. Given the very large number of associations explored, the likelihood of a type 1 error is substantial yet no correction for multiple comparison were employed. In the section on study limitations, the authors should state that given the exploratory nature of the study, no correction for multiple comparison was applied.

This limitation has now been added to the section.

Minor comments

  1. The maternal PBMCs were washed in PBS containing 2% FBS. It is highly unusual to risk contamination of human mtDNA with bovine mtDNA this way. Did the authors ascertain that the primers used to amplify the mtDNA did not also amplify bovine mtDNA?

Our sequencing technique was tested on Rho0 cells to ensure that mitochondrial pseudogenes present in the cell nucleus were not amplified and sequenced. No amplification was obtained with these negative controls. Since these Rho0 cells are grown in the presence of FBS, this precludes amplification and sequencing of bovine mtDNA. In addition, the result of an in silico search for primer matching against the mtDNA reference sequence of Bos Taurus did not support the compatibility of the primers. 

  1. Line 141. The mean ± SD number of reads…

The text has been modified accordingly

  1. Presumably children under the age of 7 have not yet decided on their gender identity. Sex not gender should be used throughout.

We indeed agree. Tables have been modified accordingly.

  1. There should be consistency on the number of decimals used in Tables (e.g. 58.5 ±4, not 10.41)

We thank the reviewer for noting these discrepancies, they have now been fixed to be more consistent throughout the text.

Thank you for helping to improve the manuscript.

Round 2

Reviewer 3 Report

Nb should be spelled out